# Apoptosis: Activation and Inhibition in Health and Disease

**DOI:** 10.3390/medsci6030054

**Published:** 2018-07-04

**Authors:** Sandra Georgina Solano-Gálvez, Jack Abadi-Chiriti, Luis Gutiérrez-Velez, Eduardo Rodríguez-Puente, Enrique Konstat-Korzenny, Diego-Abelardo Álvarez-Hernández, Giorgio Franyuti-Kelly, Laila Gutiérrez-Kobeh, Rosalino Vázquez-López

**Affiliations:** 1Departamento de Microbiología y Parasitología, Facultad de Medicina, Universidad Nacional Autónoma de México, Ciudad de México 04510, Mexico; solano-sandra@hotmail.com; 2Departamento de Microbiología, Centro de Investigación en Ciencias de la Salud, Facultad de Ciencias de la Salud, Universidad Anáhuac México Campus Norte, Huixquilucán Estado de México 52786, México; jackabadi2@gmail.com (J.A.-C.); luizgoolz@gmail.com (L.G.-V.); dr.eduardorod@hotmail.com (E.R.-P.); enriquekonstat@gmail.com (E.K.-K.); diego.alvarez.hernandez@hotmail.com (D.-A.Á.-H.); 3Medical IMPACT, Infectious Disease Department, Mexico City 53900, Estado de México, Mexico; giorgio.franyuti@gmail.com; 4Unidad de Investigación UNAM-INC, División Investigación, Facultad de Medicina, Universidad Nacional Autónoma de México, Instituto Nacional de Cardiología, Mexico City, 14080, Mexico; lgutierr@unam.mx

**Keywords:** apoptosis, cancer, cell death, infection, *Leishmania*

## Abstract

There are many types of cell death, each involving multiple and complex molecular events. Cell death can occur accidentally when exposed to extreme physical, chemical, or mechanical conditions, or it can also be regulated, which involves a genetically coded complex machinery to carry out the process. Apoptosis is an example of the latter. Apoptotic cell death can be triggered through different intracellular signalling pathways that lead to morphological changes and eventually cell death. This is a normal and biological process carried out during maturation, remodelling, growth, and development in tissues. To maintain tissue homeostasis, regulatory, and inhibitory mechanisms must control apoptosis. Paradoxically, these same pathways are utilized during infection by distinct intracellular microorganisms to evade recognition by the immune system and therefore survive, reproduce and develop. In cancer, neoplastic cells inhibit apoptosis, thus allowing their survival and increasing their capability to invade different tissues and organs. The purpose of this work is to review the generalities of the molecular mechanisms and signalling pathways involved in apoptosis induction and inhibition. Additionally, we compile the current evidence of apoptosis modulation during cancer and *Leishmania* infection as a model of apoptosis regulation by an intracellular microorganism.

## 1. Background

The term apoptosis was coined by Kerr in 1972 to specifically define a process of programmed cell death, characterized by morphologic and molecular changes distinct to every other type of cell death. The morphologic changes in the cells that occur during apoptosis include progressive rounding of the cell, retraction of pseudopods, cellular, and nuclear volume reduction (pyknosis), nuclear fragmentation (karyorrhexis), structural modification of organelles, and formation of vesicles due to blebbing of the plasma membrane [1,2]. According to the 2009 classification by the Nomenclature Committee on Cell Death (NCCD), cell death can be either programmed (i.e., apoptosis and autophagic cell death) or non-programmed (i.e., necrosis) [2]. In 2012, the NCCD proposed that apoptosis should not only be defined by morphological cell changes, but also by quantifiable biochemical parameters. For this reason, the NCCD has considered particular molecular events specific to each type of cell death. In the case of apoptosis, depending on the activation pathway, the NCCD distinguishes between extrinsic apoptosis, caspase-dependent intrinsic apoptosis, and caspase-independent intrinsic apoptosis [3,4]. Later, Galluzi and colleagues reclassified the different types of cell death in two main groups: accidental cell death (ACD) and regulated cell death (RCD) [4]. Programmed cell death (PCD) belongs to the RCD group along with apoptosis because they necessarily involve a series of biochemical processes responsible for cell death [4,5,6,7,8]. 

Apoptotic cell death and apoptosis inhibition are two phenomena that occur in health and disease, and although specific processes that occur during apoptosis have been widely studied, there is still new research and discoveries to be made. The purpose of this work was to better understand the signalling pathways of these complex and intricate processes and the pathophysiology of diseases where apoptosis is deregulated. Knowledge about the involved signalling pathways and molecules is of vital importance because they represent targets for the design of new drugs to combat pathologies where apoptosis is not properly regulated. 

## 2. Generalities of Apoptosis

The apoptotic process is composed of an intricate web of intracellular signalling pathways encompassed in three phases: initiation or activation, execution, and cellular demolition, and can be triggered in three different ways: extrinsic pathway, intrinsic pathway (subdivided in mitochondrial-induced apoptosis and endoplasmic reticulum stress-induced apoptosis) and caspase-independent pathway (Figure 1) [4,5,6,7,8].

## 3. Activation

### 3.1. Extrinsic Pathway

This pathway is activated via extracellular stress signals that are detected and disseminated by specific transmembrane receptors [9,10,11] which can be referred as lethal receptors or death receptors. These have been deeply analysed and among the better understood are tumour necrosis factor receptor (TNFR), Fas receptor (FasR), death receptor 3 (DR3), and TNF-related apoptosis-inducing ligand (TRAIL) [12,13]. Other pro-apoptotic extrinsic signals include netrin receptors such as Uncoordinated 5 A-D (Unc5A-D) and Deleted in Colorectal Cancer (DCC), which carry out their lethal functions only when their corresponding ligands fall below a critical threshold [11]. Death receptors possess intracellular domains referred to as death domains (DD), which include the TNFR1 associated death domain protein (TRADD) and Fas-associated protein with death domain (FADD) [14]. Once receptors become engaged with their respective ligands, activating proteins are recruited, such as receptor interacting protein kinase-1 (RIPK1), FADD, cellular FLICE-like inhibitory protein (c-FLIP), inhibitory PAS domain protein (c-IPA), and ubiquitin ligase E3 [15,16,17,18,19,20]. The resulting supramolecular complex, formed by the activating protein-receptor domain, is recognized as a death-inducing signalling complex (DISC), which activates procaspase-8, the precursor of caspase 8 [14,17,18,19,20,21]. Moreover, as previously mentioned, the extrinsic pathway can be triggered without a ligand, as happens with DCC and UNC5B receptors. In the absence of a ligand, DDC interacts with the cytoplasmic adapting protein named downregulated in rhabdomyosarcoma LIM-domain protein (DRAL) to assemble an activation platform for caspase-9 [22]. In a similar manner, the Unc5B receptor, in the absence of netrins, recruits a molecular complex composed of protein phosphatase 2A (PP2A) and death associated protein kinase 1 (DAPK1) [23]. In both cases, caspase-8 is activated to initiate cell death via apoptosis.

### 3.2. Activation of the Intrinsic Pathway via the Mitochondrial-Induced Apoptosis

This pathway can be activated through intracellular stimuli such as irreversible genotoxic damage, high calcium (Ca^+^) concentrations in the cytoplasm and oxidative stress. Furthermore, other mechanisms have also been described [14]. A family of proteins called Bcl-2 (characterized for having from one to four conserved domains that share homology with Bcl-2 or BH) participate in this pathway. This family is composed of proapoptotic proteins (Bax, Bak, Bad, Bcl-Xs, Bid, Bik, Bim, and Hrk) and antiapoptotic proteins (Bcl-2, Bcl-X_L_, Bcl-W, Bfl-1, and Mcl-1). The antiapoptotic proteins present the four mentioned Bcl-2 homology (BH) domains (BH1-BH4), while the proapoptotic proteins are further subdivided into Bax and “BH3-only” subfamilies, according to the BH domains that they possess [5]. The Bax subfamily is composed of Bak, Bax, Bok, and Mtd and possesses three BH domains (BH1–BH3). The BH1 and BH2 domains are structurally similar to the diphtheric toxin [24,25], but the “BH3-only” subfamily is composed of Bid, Bad, Bim, Bik, Blk, Hrk, Noxa, or Puma, and is characterized for having a single BH3 domain. Although the mechanism of action of the proapoptotic “BH3-only” subfamily is still unknown, there is evidence that points out that their action could be either performed in a direct or indirect way. In the direct way, they induce the formation of pores in the mitochondrial membrane and in the indirect way, they activate and release proapoptotic proteins to inhibit antiapoptotic proteins. The antiapoptotic proteins Bcl-2 and Bcl-xL are located in the outer mitochondrial membrane and prevent the release of cytochrome C. On the other hand, the proapoptotic proteins Bad, Bid, Bax, and Bim are located in the cytosol and under certain stimuli, they translocate to the mitochondria, where they induce the release of cytochrome C [24,25]. Additionally, caspase-8 may take part in the intrinsic pathway through Bid proteolysis, turning it into tBid, which also translocates to the mitochondria and activates Bcl-2, Bax, and Bak [26]. Once Bax and Bak have been translocated to the mitochondrial membrane, a molecular complex known as permeability transition pore complex (PTPC) is activated, which induces the mitochondrial transition permeability (MTP) phenomenon [27,28]. These series of events lead to permeability of the outer mitochondrial membrane (MOMP), which is the rate-limiting step in apoptosis, because it conducts to three lethal events: (1) loss of the mitochondrial transmembrane potential (MMP), which impedes ATP synthesis, as well as mitochondrial transport activities that depend on such potential; (2) release of toxic proteins such as cytochrome C, apoptosis-inducing factor (AIF), endonuclease G (EndoG), Smac, and HtrA2 from the inner mitochondrial membrane to the cytoplasm; (3) and inhibition of metabolic processes, such as the respiratory chain, and overproduction of reactive oxygen species (ROS) [29,30]. Once MOMP is generated, energetic and metabolic damage is produced and the cell faces irreversible apoptotic cell death. The release of cytochrome C from the mitochondria permits its association with the apoptosis protease-activating factor (Apaf-1), thus forming a structure to which procaspase-9 is incorporated, giving rise to a molecular complex referred to as the apoptosome. As procaspase-9 is activated, it recruits executor caspases-3 and 7, which leads to a proteolytic effect and induces cell death [26]. 

### 3.3. Intrinsic Pathway via Endoplasmic Reticulum Stress-Induced Apoptosis

The main stimulus of this pathway is the misfolding of proteins and their subsequent accumulation in the endoplasmic reticulum (ER) (Figure 2). Once the misfolded proteins reach a critical concentration, ER membrane sensors such as protein kinase RNA-like endoplasmic reticulum kinase (PERK), inositol-requiring protein 1 (IRE1a) and activating transcription factor 6 (ATF6) are activated. Initially, PERK phosphorylates the eukaryotic translation initiation factor 2A (EIF2A) to reduce protein transcription and translation as a regulatory feedback mechanism. Other transcription factors such as activating transcription factor 3 (ATF3), activating transcription factor 4 (ATF4), and nuclear erythroid 2-related factor (NrF2) are also inhibited by PERK [31]. On the other hand, IRE1a possesses RNAse activity and activates X-Box-binding protein-1 (XBP1), which, in turn, activates the production of ER chaperone proteins and endoplasmic reticulum-associated degradation (ERAD) products. Further, ATF6 migrates to the Golgi apparatus, where it is proteolytically activated into ATF6n, which also promotes XBP1 production. Apoptosis occurs through the association of IRE1 with Bax and Bak and at the same time the activation of the MAP kinases p38 and Jun N-terminal kinases (JNK) when these sensor mechanisms are not able to compensate the misfolded protein concentration [31].

### 3.4. Caspase Independent Pathway

Mitochondrial damage induces the release of diverse molecules with proapoptotic capability, including AIF, EndoG, and HtrA2. These three molecules by themselves can induce apoptosis without the caspases as intermediaries. AIF and EndoG can enzymatically attack DNA, while HtrA2 is capable of proteolytically attack the cytoskeleton [32]. 

## 4. Execution

### 4.1. Generalities of Caspases

Caspases are proteases that require the presence of cysteine to perform their catalytic activity. Their proteolytic function lies specifically in an aspartate residue, and hence the origin of their name (Cysteine-dependent ASPartate-specific peptidASE) [5]. Although many types of caspases exist, they are classified according to their function in the following manner: Initiation caspases (2, 8, 9, and 10); executor caspases (3, 6, and 7); and inflammatory caspases (1, 4, and 5). There are other caspases that perform diverse functions, such as caspase-11, which regulates cytokines during septic shock, caspase-12 which is associated with ER stress apoptosis, and caspase-14, which has only been isolated in embryonic tissue, specifically in keratinocytes. Caspases are found in cells as zymogens, with minimal or null enzymatic activity. These zymogens, also referred to as procaspases, possess three distinct regions: The first one is the prodomain region, located in the N-terminal end, followed by the major subunit and lastly the minor subunit, close to the C-terminal end. The length of the prodomain varies among the types of caspases, from 25 residues in effector caspases to 100–200 residues in initiator and inflammatory caspases [33]. The inflammatory caspases additionally possess an extra domain, which can be either caspase recruitment domain (CARD) or death effector domain (DED). On the other hand, zymogen activation can happen either through autoactivation or through activation by another caspase or molecule. This process occurs due to excisions in two sites of the aspartate residues, the first between the prodomain region and the major subunit, and the second between the major and minor subunits [34].

### 4.2. Formation of the Apoptosome

The activation of effector caspases only requires an intrachain excision mediated by an initiator caspase. Contrary to that, initiator caspase activation is a more intricate process, as it requires the formation of an apoptosome. In the apoptosome, initiator caspases are subject to proteolytic cleavage. The Apaf-1, an inactive monomer in nonapoptotic cells, is a molecule that mediates the formation of the apoptosome. This molecule is an adaptor multidomain protein that consists of CARD and nucleotide-binding oligomerization domain (NOD) and possesses an ATPase and a regulator domain in the C-terminal region composed of repeating WD40 (WDR) units [32]. The formation of the apoptosome is initiated by MOMP-mediated release of cytochrome C, which binds to Apaf-1 on the WDR domain, followed by the conversion of ADP into dATP/ATP in the NOD, thus forming the heptameric apoptosome [35,36,37]. Finally, procaspase-9 binds to Apaf-1 through a homotypical interaction with the CARD [38]. The apoptosome catalyses the autoproteolytic action of procaspase-9, and its active form, caspase-9, remains bound to Apaf-1 as a holoenzyme [38].

## 5. Cellular Demolition

Once the apoptosome is formed and caspase-9 is activated, a cascade of events occurs, where initation caspases (2, 8, and 10) and execution caspases (3, 6, and 7) take part in it [5,14]. The proteolytic action of the execution caspases is directed to multiple substrates of vital importance to the cell, e.g., the actin cytoskeleton activity regulator known as rho-associated protein kinase 1 (ROCK1). Proteolytic activation of ROCK1 leads to the loss of its C-terminal, subsequent phosphorylation and, thus, activation of the myosin light chain, which generates actin contraction that in turn triggers several phenomena, such as phosphatidylserine (PS) translocation, cellular rounding and retraction, as well as vesicle formation or blebbing and loss of intercellular unions due to proteolytic attack of desmosomes or other forms of cell to cell junctions. Due to the fact that the nuclear lamina is surrounded by actin filaments, caspases also generate the loss of nuclear membrane integrity, a process termed karyorrhexis, with further fragmentation of DNA and degradation of proteins associated with transcription and translation [5,39,40,41,42,43,44,45,46,47,48,49,50,51].

Another target attacked by caspases is the caspase-activated DNase (CAD), an endonuclease found inactive in healthy cells, where it forms a complex with inhibitor-CAD (ICAD) proteins. When apoptosis is initiated, caspases induce a proteolytic attack on ICAD, allowing CAD activation, and the subsequent DNA degradation at internucleosomal sites [50]. In a similar manner, when apoptosis is initiated, caspases also activate a family of proteins known as Golgi reassembly and stacking proteins (GRASP) that participate in Golgi apparatus conformation, cistern formation, and connections leading to Golgi fragmentation and disintegration [5,52].

Continuing with the demolition events, in the mitochondria, Bak and Bax proteins are activated by BH3 action, which in turn generate pores in the mitochondrial membranes and subsequent release of their contents. Additionally, the p75 subunit of the electron transport chain complex 1 is proteolytically degraded [5,52]. One of the final acts of apoptosis is the release of chemotactic cytokines and other molecules, as well as the formation of union sites for phagocytic cells quintessential for the elimination of cellular remains by phagocytes [5,53]. Some of the chemotactic signals necessary for the arrival of phagocytic cells to the site of cell death have been identified, such as lysophosphatidylcholine, sphingosine-1-phosphate, and the endothelial monocyte-activating oolypeptide II (EMAPII) [54,55]. Furthermore, fractalkine, a chemotactic molecule attached to the membrane and secreted by apoptotic cells, recruits other phagocytic cells such as microglia. Macrophages (Mφ) recognize apoptotic cells thanks to the exposure of PS in the external face of the plasma membrane, which is flipped to this side through a flippase. Once the effector caspases are activated, flippases are no longer capable of flipping the phospholipids. PS exposure also induces the release of anti-inflammatory cytokines. Other molecules such as oxidized low-density lipoproteins in the surface of apoptotic cells permit its interaction with the scavenger receptor class A (SR-A) and lectin-like oxidized low-density lipoprotein receptor-1 (LOX1) in the surface of Mφ [5,54,56].

## 6. Signal Transduction Pathways in Apoptosis

### 6.1. Mitogen-Activated Protein Kinase (MAPK) Family

Several signal transduction pathways have been implicated in the activation or prevention of apoptosis with mitogen-activated protein kinase (MAPK) playing a leading role. The MAPK family consists of protein kinases activated by mitogens and other physical and chemical stimuli, such as growth factors, ultraviolet radiation (UVR), genotoxic agents, oxidative stress, inflammatory stimulation, and cytokines. Such stimuli may produce cellular proliferation, differentiation and apoptosis [57,58]. MAPKs are characterized for having three sequential phosphorylation steps [59], carried out by three groups of enzymes: MAPK kinase kinase (MAPKKK), for example apoptosis signal-regulating kinase 1 (ASK1) and transforming growth factor-b-activated kinase 1 (TAK1); the MAPK kinase (MAPKK), for example Mitogen-activated ERK (Extracellular Signal-Regulated Kinases) kinase MEK 1 through 7); and MAPK, such as ERK 1/2, JNK, and p38. Mitogen-activated protein kinases are serine/threonine type kinases [60,61,62,63,64,65] and possess tyrosine (Tyr) and threonine (Thr) conserved double phosphorylation domains [59]. They are further divided in three subfamilies according to the amino acid present in both phosphorylation sites. (Thr-XXX-Tyr): the p38-MAPK subfamily features glycine in between two phosphorylation sites (Thr-Gly-Tyr) and is activated through stress signals, growth and differentiation factors. This subfamily is composed of the p38-MAPKα, p38-MAPKβ, p38-MAPKγ, and p38-MAPKδ isoforms; the JNK subfamily features proline between the two phosphorylation sites (Thr-Pro-Tyr) and is activated by stress signals. This subfamily is composed by the JNK1, JNK2, and JNK3 isoforms; and the ERK subfamily features glutamic acid in between two phosphorylation sites (Thr–Glu-Tyr) and is activated mainly by growth factors. This subfamily is composed of the ERK1 and ERK2 isoforms [59,60,61,66].

### 6.2. p38

This protein was identified in 1994 by Lee and colleagues [67] in lipopolysaccharide-stimulated Mφ as a tyrosine phosphorylated protein and was denominated MAPK p38. The genes that codify for the p38α, p38β, p38γ, and p38δ isoforms have been identified. Isoforms share a 12-amino acid activation loop and differ in affinity for the activating protein, tissue expression, and downstream effect. They participate in the regulation of certain growth factors, kinases, and phosphatases, as well as in the regulation of activating transcription factor 2 (ATF-2), myocyte enhancer factor (MEF2), MAPK activated protein kinase (MAPKAPK), cell division cycle 25 (CDC25), or mitogen- and stress-activated protein kinase 1 and 2 (MSK1/2). Their activation triggers cellular survival, development, and maturation [68,69,70,71,72,73,74]. The p38α isoform, commonly referred to as p38, as well as the p38β isoform, are present in almost every tissue. Contrary to that, p38γ and p38δ isoforms have a more restricted localization; the first being present in skeletal muscle and the second in the lungs, kidneys, testicles, pancreas, and small intestine [75,76]. The activation of p38 starts when stress conditions, such as genotoxic or osmotic shock activate mitogen-activated protein kinase kinase kinase 3 (MEKK3), MEK4 or TAK1 which are phosphorylated downstream into MKK3, MKK6, and very rarely MKK4, which in turn activate p38 by phosphorylating specifically at Thr180 and Tyr182 sites. This phosphorylation process produces conformational changes that lead to the enzyme binding with ATP and the acceptor substrate of the phosphate [62,63,64,65,77].

### 6.3. Jun N-Terminal Kinase (JNK)

JNK proteins are also known as stress-associated MAPKs or stress-activated protein kinases (SAPKS). They participate in cellular growth, differentiation, and apoptosis [78,79] as a response to several stress signals, such as hyperosmolarity, UVR or gamma radiation, ischemic damage, thermal shock, toxins, peroxides, protein synthesis inhibitors (anisomycin), antineoplastic drugs, and inflammatory cytokines, among others [78]. Stress signals activate TAK1, MAP3K, ASK1, and ASK2, which in turn activate MEK4 and MEK7 through phosphorylation of two specific serine (Ser) and Thr residues. MEK4 and MEK7, also known as MKK4 (SEK1/JNKK1) or MKK7 (SEK2/JNKK2) are both MAPKK, and phosphorylate JNK in Thr-Pro-Tyr specific residues [62,63,64,65,78,80,81]. JNK are codified by three genes: NK1 (46 kDa) [82,83], JNK2 (55 kDa) [84] and JNK3 (48 kDa) [85]. These genes are subject to at least ten different types of alternative splicing in order to generate the different isoforms that up to date are the following: JNK1α1, JNK1α2, JNK1β1, JNK1β2, JNK2α1, JNK2α2, JNK2β1, JNK2β2, JNK3α1, and JNK3α2. Although these isoforms are physically different, their biological roles are similar [86]. JNK1 and JNK2 isoforms are expressed in all tissues, while JNK3 isoform is found predominantly in nervous tissue, and to a lesser extent in the heart and sperm [74,87,88]. Although JNK1, JNK2, and JNK3 can all induce apoptosis, evidence suggests that each protein induces apoptosis through a different pathway. It has been demonstrated that all of them associate with p53, to activate proapoptotic gene expression, such as Bax or Puma, but interestingly, their expression varies with respect to p53. In the case of JNK1, its expression is inversely proportional to p53, contrary to JNK2 expression, which is directly proportional to p53. Both JNK2 and JNK3 can phosphorylate p53, while JNK1 can only modify it post-transcriptionally [89,90].

### 6.4. Extracellular Signal–Regulated Kinase 1/2 (ERK1/2)

ERK1 (also known as MAPK3 or p44MAPK) and ERK2 (also known as MAPK1 or p42MAPK) are kinases activated by growth factors, hormones, and neurotransmitters through binding to G-protein coupled receptors, tyrosine-kinase receptors, and ion channels [81,91]. Once bound, signal transduction continues with an adaptor protein that transmits the signal to a MAP3K, such as Raf-1B-Raf, A-Raf, and tumour progression locus 2 (TPL2). Following the described phosphorylation pattern (MAPKKK → MAPKK → MAPK), the stimulus activates MAPKKK (i.e., Raf-1), which, in turn, phosphorylates MEK1 and MEK2 and these finally phosphorylate and activate ERK1 and ERK2 [81].

## 7. Participation of MAPK in Apoptosis

One of the utmost actions of MAPK is the activation of transcription factors, which regulate gene expression and lead to crucial molecular events in the cell affecting growth, proliferation, inflammatory cytokine production, and apoptotic cell death [77]. In relation to apoptosis, a key participant is JNK that plays its role through two different mechanisms: The first one is related to nuclear events in which JNK is translocated to the nucleus and activates c-Jun and other transcription factors that promote proapoptotic gene expression, through p53/73 or c-Jun/activator protein 1 (AP-1)-dependent mechanisms [92,93]; the second one relates to JNK activation and translocation to the mitochondria, where it promotes the phosphorylation of protein 14-3-3, a protein that normally inhibits Bax. As protein 14-3-3 is phosphorylated, Bax is released and translocated to the interior of the mitochondria where it oligomerizes and forms pores in the mitochondrial membrane with the subsequent release of cytochrome C, and apoptosis induction through the intrinsic pathway. Apart from these two mechanisms, JNK can also phosphorylate “BH3-only” family members, whose antiapoptotic effect inhibits Bcl-2 and Bcl-xL and is also involved in the posttranslational modifications of Bid and Bim, both of which induce Bad and Bax activity [92,93]. Another MAPK deeply involved in apoptosis is p38, which may be simultaneously activated with JNK [94]. P38 exerts its central role in apoptosis through the activation of proapoptotic proteins, mainly Bad, Bax and Bim Extra-Long (BimEL), [95,96,97,98,99] and simultaneously induces the inhibition of ERK and Akt antiapoptotic pathways [98,99]. Additionally, p38 and JNK participate in Toll-like receptor (TLR) signalling pathways. These key participants of the innate immune response function as regulatory sensors of both apoptosis signalling through the induction of MAPK p38 and JNK [100,101] and survival signals through Phosphatidylinositol 3-Kinase (PI3K) and some Bcl-2 family members in dendritic cells (DC) [102,103,104]. 

## 8. Transforming Growth Factor-β-Activated Kinase 1 (TAK1) and Apoptosis

Transforming growth factor β activated kinase (TAK1) was initially identified as a mitogen-activated kinase kinase kinase (MAP-K3) and, to date, has been found to be activated by a wide variety of stimuli that include TGF-β, bone morphologic protein, other cytokines, such as TNF- β and IL-1, Toll-like, B-cell, T-cell, death receptors–ligands, and ceramide. Environmental changes and exogenous stressors also can activate it. TAK1 has a fundamental pro-survival function and also has been found to participate actively in the RIPK-1 and RIPK-3-mediated necroptosis. The activation route of TAK1 upon TNF- β stimulation has been extensively studied and involves the recruitment of different molecules such as TRADD, TRAF-2 and 5, cIAP1/2, and RIPK1. These molecules along with TNFR1 form complex I in which RIPK1 acquires a polyubiquitin chain to which TAK1 binds through TAK1 binding protein 2 (TAB2), and activates the IKK complex, leading to the activation of NF-κB. TAK1 also activates MAPK cascades. Either route of TAK1 activation leads to the expression of inflammatory cytokines and antiapoptotic proteins. Under some circumstances, after complex I formation, a dissociation of TNFR1 may exist leading to the formation of cytosolic protein complex known as complex IIa composed of TRADD, FADD, RIPK1, and caspase-8. Through this route caspase-8 activation initiates a caspase cascade, which leads to apoptotic cell death. The dual role of TAK1 in cell survival and cell death and the fact that the dysregulation of the signalling pathways that activate it in mice leads to tissue abnormalities, make TAK1 very relevant molecule regarding disease pathogenesis [62,63,64,65].

## 9. Phosphatidylinositol 3-Kinase (PI3K)/Akt Signalling Pathway

MAPK, p38 and JNK play an important role in apoptosis induction while PI3K activation promotes cellular survival. PI3K is a heterodimer formed by a p85 regulatory subunit and a p110 catalytic subunit responsible for phosphate transfer. The signalling pathway initiated by this kinase is activated by different stimuli, growth factors standing out among them. Once a ligand binds to the Tyr specific Tyr-kinase receptor, an insulin receptor substrate (IRS) adaptor protein is activated, which in turn activates the regulatory PI3K subunit and generates a conformational change that allows the binding of the catalytic subunit and thus the assembly of the active molecule that catalyses the conversion of phosphatidylinositol 4,5-biphosphate (PIP_2_) into phosphatidylinositol 3,4,5-Triphosphate (PIP_3_) [72,105]. PIP_3_ interacts with the pleckstrine homology (PH) domain, located in the N-terminal region of the Ser/Thr kinase Akt or protein kinase B (PKB), resulting in the recruitment of the kinase to the plasma membrane [106,107,108]. Further, phosphoinositide dependent kinase 1 (PDK1) phosphorylates Akt/PKB producing a conformational change that facilitates a second phosphorylation by the rapamycin-intensive companion of mammalian target of rapamycin (RICTOR)-mammalian target of rapamycin complex 1 (mTOR1) [109]. Finally, the PI3K/Akt pathway leads to diverse effects associated with cellular proliferation and survival [110,111]. Specifically, it produces the inactivation of many proapoptotic signals, such as Bad, procaspase-9, and Forkhead (FKHR) transcription factors [112,113]. It also promotes the activation of cyclic AMP response element binding (CREB) protein, nuclear factor κB (NF-κB), and hypoxia-inducible factor 1-alpha (HIF-1α), which in turn activate the expression of antiapoptotic genes [114,115,116]. 

## 10. Apoptosis in Physiological Processes

The word apoptosis has its etymological origin in the Greek *apó*, which means “*from*” and *ptōsis* which means “*falling off*”. Merging these two words is an allusion to the natural events of shedding cells and tissues, as well as the falling of old leaves during autumn. The word apoptosis describes the process in which unwanted, damaged, or old cells are eliminated in multicellular organisms [117]. It is a necessary process in all body tissues and happens naturally during embryogenesis, metamorphosis and constant cellular changes, being of utmost importance for the maintenance of homeostasis in all tissues [118]. Such is the case of the constant cellular changes that occur in the skin. Diverse biochemical and structural analysis have demonstrated that apoptosis is a normal process of keratinocytes in the epidermis and radicular sheath [118,119]. The skin in general, but more specifically the epidermis, is exposed to diverse factors that induce apoptosis, including UVR, oxidative stress, cytokines, chemokines, cytotoxic T cells, and Mφ, among others. Therefore, defence mechanisms are required to maintain its integrity and capacity to replenish cells of the epithelium and annexes [120,121]. In normal keratinocytes, programmed cell death begins in the stratum granulosum of the epidermis. The Bcl-2 gene codifies an antiapoptotic protein and is expressed in the basal layers of keratinocytes. However, its expression decreases in the suprabasal layers, which has been associated with the differentiation process undergone by the cells present in these layers [122]. These cells then migrate from the suprabasal strata to superficial layers, probably by the inhibition of Bcl-2. Additionally, suprabasal keratinocytes stop proliferating as a normal response to the low expression of the c-myc oncogene or the decreased synthesis of tumour growth factor 13 (TGF-13) [122]. Apoptosis has also an outstanding role in the regression of the corpus luteum, a process termed luteolysis. The ovulatory process is ensued by the transformation of the remaining follicle into the corpus luteum, a structure that secretes progesterone. If fertilization does not occur, the lack of production of human chorionic gonadotropin (hCG) causes the involution of the corpus luteum where apoptosis is initiated via TNF, Fas/FasL, and caspase-3, and alters the equilibrium of Bcl-2/Bax expression, through a similar mechanism as the one previously described for skin [123,124]. The alteration in the equilibrium between cell death and survival can also lead to different pathological processes as has been demonstrated with certain intracellular infections, as well as neoplastic processes where apoptosis is inhibited. Contrary to this, in certain invasive infections and autoimmune diseases, an overactivation of cell death can occur [118].

## 11. Apoptosis Inhibition and Infection

The main type of programmed cell death observed in infections is apoptosis [125,126,127]. Apoptosis inhibition is a resource to which many intracellular organisms such as bacteria, virus, fungi, and parasites recur. By doing this, microorganisms achieve persistence in cells to obtain nutrients, reproduce, and avoid being recognized by the immune system. A thoroughly studied example of inhibition of apoptosis by an intracellular pathogen is the one caused by *Leishmania* (Figure 3).

### 11.1. Leishmania Participation in Apoptosis Inhibition

*Leishmania* is an obligate intracellular protozoan parasite with high metabolic dependence on parasitized cells. This parasite may infect a variety of cells, but Mφ and DC are arguably the most important cells where *Leishmania* survives and replicates to maintain infection [125]. To achieve this feature, *Leishmania* must downregulate or inhibit different defence mechanisms of host cells. Inhibition of apoptosis is one of the most important of them. Studies have demonstrated that monocytes, DC and Mφ grown in apoptogenic conditions, however, if they become infected with *Leishmania* sp. they do not go into apoptosis. For example, it has been demonstrated that infection with *L. donovani*, or LPG stimulus, inhibits apoptosis in Mφ, and owing to cellular activation, the production of TNF-α, TGF-β, interleukin-6 (IL-6), and granulocyte-macrophage colony stimulating factor (GM-CSF) increase, while secretion of M-CSF and IL-1β decrease [128]. It has also been demonstrated that *L. major* inhibits the release of mitochondrial cytochrome C in infected Mφ grown in the presence of staurosporine, thus delaying apoptosis [129]. Other studies have reported similar results, as in the case of the monocyte cell line U937 infected with *L. infantum* where inhibition of actinomycin D-induced apoptosis was observed [130], or in Mφ from the cell line RAW 264.7 infected with *L. major* where apoptosis diminished even with the presence of cycloheximide [131]. In infected neutrophils, spontaneous apoptosis was inhibited by *L. major* due to a decrease in caspase-3 activity [132]. It has also been demonstrated that amastigotes and promastigotes of *Leishmania mexicana* inhibit camptothecin-induced apoptosis in monocyte-derived Dendritic Cells (moDC) [133,134].

### 11.2. Role of *Leishmania* in the Modulation of MAPK and PI3K

*Leishmania* has the capacity to inhibit apoptosis of different cells, however, the mechanisms involved in this inhibition have not been fully understood. Regarding the role of *Leishmania* infection in the modulation of proapoptotic pathways such as MAPK, it has been shown that *L. mexicana* amastigotes and promastigotes significantly reduced MAPK, JNK, and p38 phosphorylation in moDC [135,136]. The inhibitory effect was only observable in immature DC because maturation driven by the stimulation with lipopolysaccharide (LPS) did not suppress MAPK phosphorylation [137]. In bone marrow macrophages (BMM) previously stimulated with interferon gamma (IFN-γ), it was also shown that *L. donovani* promastigotes exerted a similar effect inhibiting the activation of p38, JNK, and ERK that was directly associated with TNF-α production and ensured the survival of the parasite [138]. Other authors also demonstrated that inhibition of p38 was associated with an increase in the number of infected Mφ and parasite survival [139]. Interestingly, not only the parasite but also some surface components such as gp63 have been shown to inhibit the apoptotic signalling of MAPK p38 [140]. On the other hand, *Leishmania* infection can also activate MAPK. Such is the case of BMM infected with *L. amazonensis* where it has been observed that ERK 1/2 activation generates an epigenetic modification in the IL-10 locus, which results in a great induction of this cytokine in infected Mφ. The modification is the result of histone H3 and IL-10 promoter phosphorylation, which allows the binding of the IL-10 promoter with Sp1 transcription factor [141]. In addition, Mφ grown in presence of LPG showed an altered production of IL-12 associated with ERK activation and signalling [142]. Other authors demonstrated that ERK 1/2 activation induced by *L. amazonensis* yielded a lesser expression of CD40 and IL-12 production in Bone marrow derived dendritic cells (BMDC), with the subsequent inhibition of DC maturation. Otherwise, specific ERK 1/2 inhibition induced the production of Nitric Oxide (NO) which caused an increase in parasite death [143].

Interestingly, *Leishmania* infection not only intervenes with signalling pathways that induce apoptosis, but also with pathways that promote survival as it has been shown with the infection of BMM with *L. major* and *L. pifanoi* promastigotes that promote resistance to apoptosis through activation of PI3K/Akt. It was also demonstrated that Akt phosphorylates Bad, which, in turn, interacts with the 14-3-3 protein, inhibiting it and boosting the antiapoptotic action of Bcl-2 [144]. It has also been demonstrated that infection of moDC with *L. mexicana* amastigotes activated antiapoptotic signals, such as PI3K/Akt phosphorylation [136]. Recently, the activation of two PI3K isoforms, PI3Kγ (ROS dependent) and PI3Kδ (ROS independent) was demonstrated in neutrophils infected with *L. amazonensis*. These isoforms, in turn, activate the ERK pathway downstream, which is associated with the process of netosis and subsequent activation of ROS and release of neutrophil extracellular traps (NET) [145]. 

## 12. Apoptosis Inhibition and Cancer

The process of tumorigenesis is complex, characterized among other things for a newly-acquired capacity of immune system evasion, as well as implantation and propagation of cancer cells. In order to achieve this big enterprise, several mechanisms are exacerbated in cancer cells such as oxidative stress, gene mutations, epigenetic changes, and overproduction of inflammatory cytokines. Additionally, neoplastic cells must activate mechanisms that inhibit cell death and apoptosis (Figure 4) [146].

### 12.1. TGF-β, Apoptosis Inhibition and Cancer

The signalling pathway of the TGF-β has a dual capacity: it can either suppress or promote neoplastic cell growth. It has been described that in both healthy cells and early-stage cancer cells, TGF-β can induce cell-cycle arrest and apoptosis, thus having tumour-suppressor functions. However, there is evidence that proves that in late-stage cancer cells, TGF-β can promote tumorigenesis, metastasis, and chemoresistance [147]. This signalling pathway starts with TGF-β binding to the transforming-growth-factor-β Receptor (TGF-βR). TGF-βR possesses two isoforms, TGF-βRI and TGF-βRII. The binding of TGF-β to TGF-βR1 causes phosphorylation of Ser and Thr residues [148], which in turn activate the adaptor proteins Smad2 and Smad3, forming the R-Smad2 and R-Smad3 complexes [149]. After this process is complete, the recently formed complex binds to Smad4, which is then translocated to the nucleus and regulates gene expression [150]. In early-stage cancer cells, TGF-β can induce apoptosis in transforming cells, however, the events that lead to tumorigenesis cause an attenuation of the cellular response towards these molecules, while TGF-β activation is increased. This, in turn, produces an overactivation of proinflammatory cytokine expression [151]. In 2003, Song and colleagues demonstrated that prostate cancer cells inhibit TGF-βRI through Akt, thus blocking the activation of Smad3. Additionally, during this survival promotion process PI3K and phosphatase and tensin homolog (PTEN) participate [152]. 

Two other members of the TGF-β family are bone morphogenic proteins (BMP) and activins. Both are able to promote cancer cell development, as well as induction of angiogenesis and epithelial-to-mesenchymal transition (EMT). Loomans and Andl proposed that through binding to Activin Receptor type IB (ActRIB), activin A induces the activation of MAPK signalling, specifically ERK 1/2, promoting cellular survival [153].

### 12.2. Ciclooxygenase 2 (COX-2) Apoptosis Inhibition and Cancer

Ciclooxygenase (COX) is the enzyme that catalyses the transformation of arachidonic acid into prostaglandins, prostacyclins, and thromboxanes, responsible for the inflammatory response. Two isoforms have been found, COX-1 and COX-2, while COX-3 is encoded by the same gene as COX-1 but has not shown the usual COX activity. COX-1 is a constitutive and homeostatic enzyme found in the cell cytoplasm while COX-2 is found in the perinuclear membrane of the cell in the gastrointestinal, renal, and musculoskeletal systems, as well as in central nervous system, ovaries, breast, and lungs [154]. It is an inducible enzyme both in physiological processes such as the arachidonic acid pathway, biosynthesis of eicosanoids, prostaglandins, thromboxanes, lipoxins, and leukotrienes as well as in pathological situations [155]. Evidence shows that in neoplastic tissues COX-2 is overexpressed, inducing an excessive production of prostaglandins with the subsequent increase of tumorigenic factors. This is exemplified by the prostaglandin-induced integrin-dependent vascular endothelial growth factor (VEGF) production, which stimulates angiogenesis, an important process for implantation, maintenance, development, survival, and metastasis of cancer cells [156,157]. Furthermore, it has also been demonstrated that COX-2 induces PI3K/Akt and ERK overexpression, promoting cancer cell survival signalling [156,157].

## 13. Discussion

Both apoptosis and its inhibition are fundamental biological processes for the homeostasis of an organism. Both processes are present throughout life and are essential for growth, development, and reproduction. Studies on the molecular mechanisms that elicit or inhibit apoptosis have been carried out in order to describe the specific signalling pathways that take place during apoptosis induction and inhibition. To this date, various routes implicated in apoptosis activation or inhibition have been studied, however, there is still much to be found. These complex and meticulous pathways are thought to be unidirectional, but it might be better to think of them as circuits of intracellular signalling. Ironically, the same pathways that are involved in homeostasis and health participate in cell death processes that occur during cancer or infections. The better understanding and gaining of knowledge on these intracellular circuits and the physiopathology behind them will permit the development of new strategies and drugs to effectively treat the pertaining diseases mentioned in this work. 

## Figures and Tables

**Figure 1 medsci-06-00054-f001:**
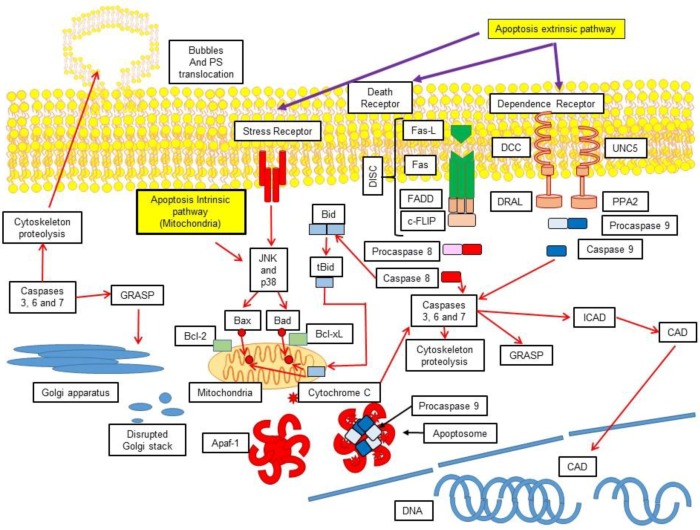
Apoptosis extrinsic and intrinsic (mitochondrial) pathways. Schematic representation of the extrinsic activation pathway and the mitochondrial intrinsic pathway. The extrinsic pathway can be activated through transmembrane death receptors (i.e., Fas, TRAIL, etc.) that detect extracellular stress signals or through dependence receptors (i.e. Uncoordinated 5 A-D (Unc5A-D), Deleted in Colorectal Cancer (DCC) which are activated when the correspondent ligand concentration is below a critical threshold; the intrinsic pathway is activated through intracellular stimuli, such as genotoxic damage, elevated calcium concentration or oxidative stress. The B-cell lymphoma 2 (Bcl-2) proteins participate in this pathway.

**Figure 2 medsci-06-00054-f002:**
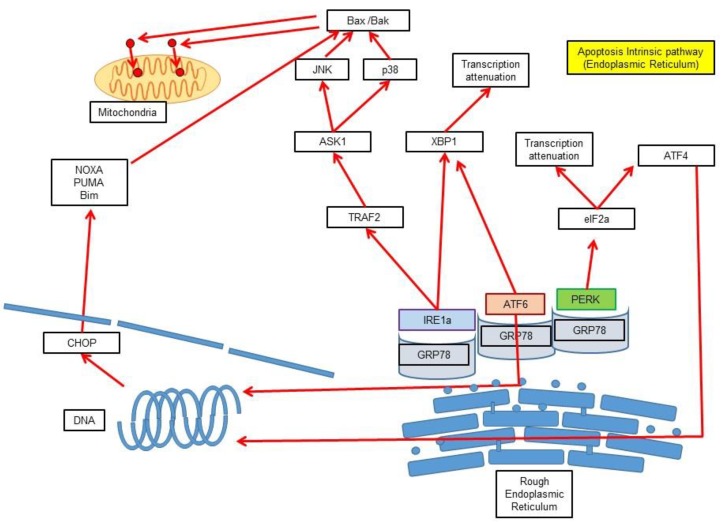
Apoptosis intrinsic (endoplasmic reticulum (ER)) pathway. Schematic representation of the intrinsic pathway of apoptosis activation in the the endoplasmic reticulum. The main stimulus for this pathway is the misfolding of proteins and their subsequent accumulation in the endoplasmic reticulum. When the misfolded proteins reach a critical concentration, ER-located sensors (protein kinase RNA-like endoplasmic reticulum kinase (PERK), inositol-requiring protein 1 (IRE1a) and activating transcription factor 6 (ATF6)) will induce proteolytic degradation and a decrease in protein synthesis.

**Figure 3 medsci-06-00054-f003:**
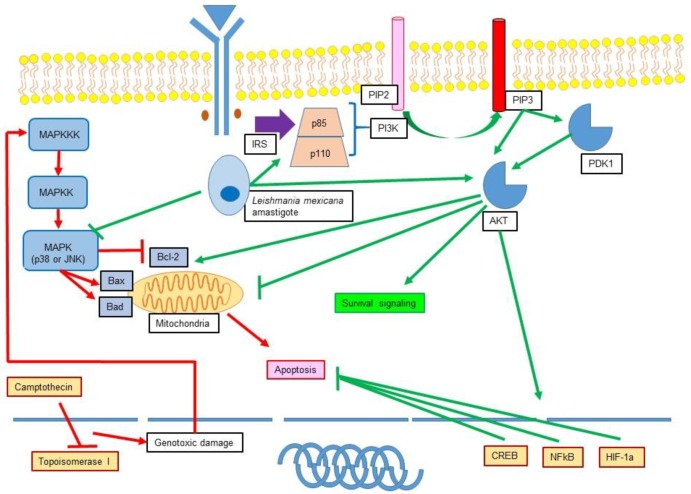
Apoptosis inhibition in *Leishmania mexicana*. Schematic representation of the proposed apoptosis inhibition mechanisms of *L. mexicana* amastigotes in human monocyte-derived dendritic cells (moDC) by Vázquez-López and colleagues. In this model, *L. mexicana* inhibits apoptosis even when dendritic cells have received a known apoptogenic stimulus (camptothecin). The parasite inhibits mitogen-activated protein kinase (MAPK), Jun N-terminal kinase (JNK) and p38 phosphorilation (proapoptotic mechanisms) and activates phosphatidylinositol 3-kinase (PI3K)/Akt (antiapoptotic mechanisms). IRS: insulin receptor substrate; PIP_2_: phosphatidylinositol 4,5-biphosphate. PIP_3_: phosphatidylinositol 3,4,5-Triphosphate. CREB: cyclic AMP response element binding protein. HIF-1α: hypoxia-inducible factor 1-alpha. NF-κB: nuclear factor kappa-light-chain-enhancer of activated B cells. PDK1: phosphoinositide dependent kinase 1.

**Figure 4 medsci-06-00054-f004:**
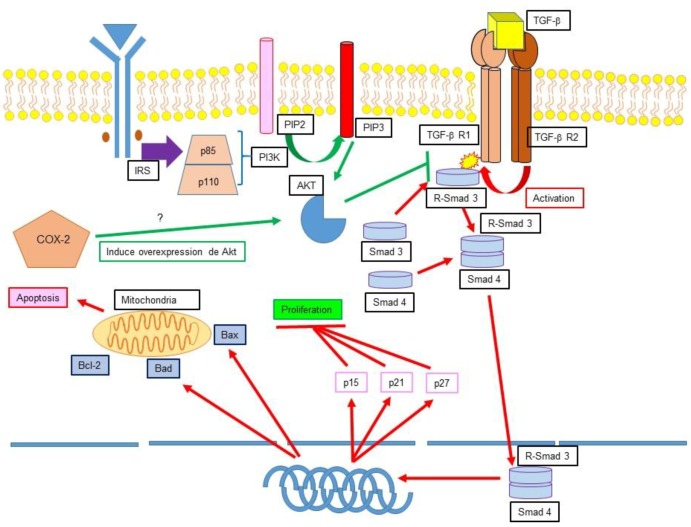
Apoptosis inhibition in cancer. Schematic representation compiling some of the described mechanisms of apoptosis inhibition in cancer cells. Certain cancer cells have been observed to activate the PI3K-Akt pathway as survival mechanisms. In healthy cells and early-stage cancer cells, Transforming growth factor β (TGF-β) can induce cell-cycle arrest and apoptosis. However, there is evidence proving that in late-stage cancer cells, TGF-β can promote tumorigenesis, metastasis, and chemoresistance. Another described mechanism for cancer cell survival is the activation of the Ciclooxygenase 2 (COX-2) pathway, which activated Akt and inhibits the overall antitumor effect of TGF-β in healthy cells or early-stage cancer cells.

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
