# Peer review of "Apoptosis: Activation and Inhibition in Health and Disease"

_medsci, 2018, doi:10.3390/medsci6030054_

Round 1
Reviewer 1 Report
Solano-galvez nicely summarized the generalities of the molecular mechanisms and signaling pathways involved in apoptosis induction and inhibition. However, in order to further strengthen the review manuscript, we recommend adding some more publications listed below.
In the “Mitogen-Activated Protein Kinase (MAPK) family” section, information regarding TAK1 MAPKKK is missing.
- Jiefei Geng et al., Nat Comm. 2017
- September R. Mihaly et al., Sci Rep. 2017, Cell Death Differ. 2014
- Antonia Sassmann-Schweda et al., JCI insight 2016
- Sho Morioka et al., J. Cell. Biol. 2014
Author Response
Comments and Suggestions for Authors
Solano-galvez nicely summarized the generalities of the molecular mechanisms and signaling pathways involved in apoptosis induction and inhibition. However, in order to further strengthen the review manuscript, we recommend adding some more publications listed below.
In the “Mitogen-Activated Protein Kinase (MAPK) family” section, information regarding TAK1 MAPKKK is missing.
- Jiefei Geng et al., Nat Comm. 2017
- September R. Mihaly et al., Sci Rep. 2017, Cell Death Differ. 2014
- Antonia Sassmann-Schweda et al., JCI insight 2016
- Sho Morioka et al., J. Cell. Biol. 2014
Response : Suggested references were included both in the bibliography and in the text. TAK1 was included as a MAPKKK that participates in the activation of MAPK p38 and JNK.
Reviewer 2 Report
The manuscript 'Apoptosis: Activation and inhibition in health and disease" by Solano-Galvez and coworkers is a well written and organized review, which covers a lot of information with regard to apoptotic signalling. This manuscript is a good summary of many signalling pathways during apoptosis.
Major concerns:
1. The authors should, however, concentrate a little bit more on the details and composition of their Figures, i.e., it would be desired if arrows are not overlapping with boxes, associated elements are aligned, names are centred in the boxes, names in Figure 3 are written with a bigger font, etc.
2. Update the introduction and include nomenclature from the last publication about recommendations of the Nomenclature Committee on Cell Death 2018 (Cell Death Differ. 2018 Mar;25(3):486-541. doi: 10.1038/s41418-017-0012-4. Epub 2018 Jan 23.). Rewrite the introduction by addition of additional recommendations from 2018.
Minor concerns:
3. Do not write about autophagy in the Abstract since it is not the main point of the review.
4. Change the name "autophagy" in the introduction to "autophagic cell death", since autophagy per se is a recovery and protective mechanism and only excessive execution of this pathway will lead to so-called "autophagic cell death". It would be worth mentioning that autophagy is not a cell death pathway but "autophagic cell death" is - see recommendations of the NCCD from the years 2009 and 2012.
5. Update the style of writing for all microorganisms in the manuscript, i.e., write all names of the microorganisms in itallics, e.g., L.donovani. Please, update throughout the whole manuscript, especially lines 379-431.
6. Update the style of writing for the Leishmania species to: "Leishmania sp." (use italics and the abbrevation "sp.")
Author Response
Major concerns:
1. The authors should, however, concentrate a little bit more on the details and composition of their Figures, i.e., it would be desired if arrows are not overlapping with boxes, associated elements are aligned, names are centred in the boxes, names in Figure 3 are written with a bigger font, etc.
Response 1: The corrections were made
2. Update the introduction and include nomenclature from the last publication about recommendations of the Nomenclature Committee on Cell Death 2018 (Cell Death Differ. 2018 Mar;25(3):486-541. doi: 10.1038/s41418-017-0012-4. Epub 2018 Jan 23.). Rewrite the introduction by addition of additional recommendations from 2018.
Response 2: The update was made
Minor concerns:
3. Do not write about autophagy in the Abstract since it is not the main point of the review.
Response 3: The word autophagy was eliminated
4. Change the name "autophagy" in the introduction to "autophagic cell death", since autophagy per se is a recovery and protective mechanism and only excessive execution of this pathway will lead to so-called "autophagic cell death". It would be worth mentioning that autophagy is not a cell death pathway but "autophagic cell death" is - see recommendations of the NCCD from the years 2009 and 2012.
Response 3: The correction was eliminated
5. Update the style of writing for all microorganisms in the manuscript, i.e., write all names of the microorganisms in itallics, e.g., L.donovani. Please, update throughout the whole manuscript, especially lines 379-431.
Response 2: The update was made
6. Update the style of writing for the Leishmania species to: "Leishmania sp." (use italics and the abbrevation "sp.")
Response 2: The update was made
Round 2
Reviewer 1 Report
The response is overall great, however, TAK1 is not merely upstream of p38 and JNK to regulate cell death, but actively participating caspase and RIPK1 dependent apoptosis in a either indirect or direct manner. Please read through the references we provided carefully and describe more about TAK1 and cell death regulation. In this section, the author should cover major TAK1-cell death papers more comprehensively.
Author Response
Comments and Suggestions for Authors
The response is overall great, however, TAK1 is not merely upstream of p38 and JNK to regulate cell death, but actively participating caspase and RIPK1 dependent apoptosis in a either indirect or direct manner. Please read through the references we provided carefully and describe more about TAK1 and cell death regulation. In this section, the author should cover major TAK1-cell death papers more comprehensively.
Response
The role played by TAK1 in caspases and RIPK1 dependent apoptosis, has been included in a separate section of the manuscript